# Validation of a Cytological Classification System for the Rapid On-Site Evaluation (Rose) of Pulmonary and Mediastinal Needle Aspirates

**DOI:** 10.3390/diagnostics12112777

**Published:** 2022-11-14

**Authors:** Lina Zuccatosta, Giulio Rossi, Stefano Gasparini, Maurizio Ferretti, Federico Mei, Michele Sediari, Francesca Barbisan, Gaia Goteri, Giuseppe Maria Corbo, Alessandro Di Marco Berardino

**Affiliations:** 1Pulmonary Diseases Unit, Azienda “Ospedali Riuniti”, 60126 Ancona, Italy; 2Pathology Unit, Fondazione Poliambulanza Hospital Institute, 25124 Brescia, Italy; 3Department of Biomedical Sciences and Public Health, Polytechnic University of Marche Region, 60126 Ancona, Italy; 4Pathological Anatomy Institute, Polytechnic University of Marche Region, 60126 Ancona, Italy; 5IRCCS, Fondazione Policlinico Universitario A. Gemelli, 00168 Roma, Italy

**Keywords:** ROSE, bronchoscopy, transthoracic needle aspiration, EBUS-TBNA, cytology

## Abstract

Rapid on-site evaluation (ROSE) is a procedure that allows immediate assessment of adequacy of cytological specimens obtained by fine needle aspiration (FNA). The application of ROSE diagnostic categories has been applied in various organs, but not in thoracic pathology. We aimed to retrospectively assess the concordance with the final diagnosis of a categorization from C1 (inadequate) to C5 (neoplastic) during ROSE performed with bronchoscopic or percutaneous sampling procedures of thoracic lesions in a large series of consecutive cases. This retrospective single-center study evaluated 2282 consecutive ROSEs performed on 1827 patients from January 2016 to December 2020 in 994 cases of transbronchial needle aspiration (TBNA) in peripheral pulmonary lesions, in 898 transthoracic FNAs, in 318 ultrasound-guided TBNAs, in 50 conventional TBNAs and in 22 endobronchial TBNAs. False positive and false negative cases of ROSE were 43 (1.88%) and 73 (3.2%), respectively, when compared with the definitive diagnosis. The sensitivity, specificity and the positive and negative prognostic values of ROSE were 94.84%, 95.05%, 96.89% and 91.87%, respectively. Overall concordance between ROSE and the final diagnosis was 0.8960 (Cohen’s kappa). No significant differences were observed in terms of sampling procedures and type and location of the lesions. A tiered classification scheme of ROSE from C1 to C5 during bronchoscopic and percutaneous sampling procedures is helpful in effectively guiding clinical management of patients with thoracic lesions.

## 1. Introduction

Rapid on-site evaluation (ROSE) is a procedure that allows assessment of adequacy of cytological specimens generally obtained by needle aspiration techniques [1,2]. Using a quick staining method, it is possible to have within few minutes a smeared cytological slide ready to be evaluated using light microscopy [3].

ROSE may be performed during a cytology biopsy from any anatomic site. In the field of respiratory medicine, ROSE is commonly used during bronchoscopic sampling techniques (transbronchial fine needle aspiration, TBNA) or transthoracic fine needle aspiration (TTNA) of lung and/or mediastinum [4,5,6,7,8,9,10,11,12,13,14].

Even if some trials failed to find a significant difference in the diagnostic yield of conventional transbronchial needle aspiration (cTBNA) or endobronchial ultrasound-guided TBNA (EBUS-TBNA) with the use of ROSE [15,16], other studies showed that ROSE determines a significant reduction of additional procedures, complications and procedure time, thus helping in obtaining sample adequacy for lung cancer diagnosis and molecular characterization of predictive biomarkers [4,5,6,7,17].

In 1999 and with the aim of succinctly and uniformly transmitting pathologic diagnostic information in a standardized format to the clinicians, the Papanicolaou Society of Cytopathology suggested classifying the ROSE results in the following five categories [3]: (1) nondiagnostic specimens; (2) specific benign lesion; (3) atypical cell present but probably benign; (4) atypical and suspicious for malignancy; (5) malignancy present.

Our group previously adopted the aforementioned tiered diagnostic scheme to evidence an 81% overall agreement between an experienced cytopathologist and a trained pulmonologist in assessing adequacy on 362 TBNAs from 84 patients with hilar/mediastinal lymphadenopathies [18].

While a similar classification has been widely used in several pathology fields such as breast, thyroid, pancreas and salivary glands [19,20,21,22], to our knowledge there are no experiences reporting the value of a tiered classification scheme (from C1 to C5) in thoracic pathology. 

Starting from 2000, the classification into five categories has been routinely adopted in our institution during ROSE performed for transbronchial needle aspiration (EBUS-TBNA and cTBNA) and TTNA for peripheral lung and mediastinal lesions.

The primary aim of the current observational, retrospective study is to validate the proposed C1–C5 classification during ROSE by comparing intraoperative results with the definitive diagnosis. Secondary outcomes are to assess sensibility, sensitivity, positive and negative prognostic values of ROSE and to evaluate whether the concordance between ROSE and the final diagnosis varies in different lesion locations (lung parenchyma vs mediastinum) and when using different sampling techniques.

## 2. Material and Methods

All transbronchial and percutaneous needle aspirations performed in the Interventional Pulmonology Unit of Azienda Ospedaliero-Universitaria, Ospedali Riuniti, Ancona (Italy) from 1 January 2016, to 31 December 2020, which underwent ROSE by an expert cytopathologist (MF or FB), were included in the study. 

The bronchoscopic procedures were performed under conscious sedation or general anaesthesia (Bronchoscopes Olympus BF-H1100, BF-H190; ultrasound bronchoscope Olympus BF-UC190F) by 5 expert bronchoscopists (LZ, SG, FM, MS and ADMB). 

For the cTBNAs, a 19 G flexible needle was used (NA-601D-1519), while for the EBUS-TBNAs, a 22 G needle (ViziShot) was employed. For the transbronchial approach to peripheral lesion, a flexible transbronchial needle (Olympus NAC1, 21 G) was used. TTNAs were performed under fluoroscopic guidance or CT scan (smaller lesions not visible at fluoroscopy) using a 22 G Chiba needle.

Immediately after the needle aspiration, the sample was flushed and smeared on a slide, fixed in alcohol and then stained using the Haemacolor Merk rapid stain system. The slides were evaluated on site by a cytopathologist and classified in the C1–C5 categories, as previously reported [18].

After the first needle aspirate, we performed three further samples to be included in 5% buffered-formalin fixative for cell block preparation. The evaluation of diagnostic yield and of the value of cell block addition is out of the aims of the present study. 

Descriptive data of population characteristics, techniques and lesions are presented as mean values and standard deviation. Sensitivity, specificity and positive and negative prognostic values of ROSE were calculated, considering the cytological final diagnosis as the gold standard. A Cohen’s kappa was used for evaluating concordance among the C1–C5 classification of ROSE and the final diagnosis of the sample as a whole and, separately, for the kind of procedure and for the location of the lesion.

The study was conducted in agreement with the Declaration of Helsinki in its latest version. The study design and protocol were previously approved by the local ethics authority (Ethic Committee of the Marche Region, approval number 209457). Due to the retrospective nature of the study and since data were anonymized, the need for informed consent was waived.

## 3. Results

The original series here included 2318 FNAs with ROSE consecutively performed in a tertiary unit of pulmonology from 1 January 2016, to 31 December 2020. After overall collection, 36 cases were excluded due to lack of complete data, and 2282 procedures on 1827 patients (1242 men, 67.9%) with a mean of age at diagnosis of 66.83 yrs ± 12.05 were evaluated in the study.

In particular, ROSE was performed using TBNA on parenchymal peripheral lesions under fluoroscopic guidance in 994 cases, while 898 TTNAs (822 on lung lesions and 76 on anterior mediastinal masses), 318 EBUS-TBNAs on mediastinal lymph nodes, 50 conventional TBNAs on mediastinal lymph nodes and 22 TBNAs on visible central endobronchial lesions were also included (Table 1). 

The mean diameter (±SD) of the targeted lesions obtained during preliminary computed-tomography (CT) scans was 30.3 mm (±16.3) for pulmonary peripheral lesions, 25.9 mm (±16.9) for mediastinal lymph nodes and 67.3 mm (±33.1 mm) for masses of anterior mediastinum.

The definitive diagnosis obtained on cytological samples is reported in Table 2. The most common diagnosis is represented by lung adenocarcinoma (24.8%), while 526 (23.05%) of the samples were not diagnostic at final evaluation.

When compared with the final diagnosis, the false positive cases were 43 (1.88%), and the false negative cases were 73 (3.2%) using ROSE. The sensitivity, specificity and the positive and negative prognostic values of ROSE were 94.84%, 95.05%, 96.89% and 91.87%, respectively. 

Figure 1 represents the diagnostic flow, the number and the percentage of C1–C5 categories for each procedure.

The distribution of the tiered “C1–C5 classification” scheme used during ROSE and definitive diagnosis is reported in Table 3 and Figure 2.

In the case of a nondiagnostic ROSE (C1) or indeterminate examination (C3–C4), the sample was repeated once in 360 patients and twice in 100 patients.

Concordance between ROSE and the final diagnosis was 0.8960 (Cohen’s kappa). 

Finally, the concordance according to the procedures and the specific site of the lesions was reported in Table 4 and Table 5, although no significant differences were observed at statistical analysis. 

## 4. Discussion

ROSE is a widely used and effective diagnostic procedure during a needle aspiration biopsy of different organs [19,20,21,22,23,24,25]. In the field of thoracic pathologies, several studies have demonstrated the utility of ROSE during both transbronchial and percutaneous needle aspiration techniques [1,4,5,6,7,8,9,10,11,12,13,14]. Even if some works failed to demonstrate that ROSE increases the sensitivity during cTBNA or EBUS-TBNA [15,16], there is robust evidence that ROSE may significantly reduce the need for additional passes and the complication rate, most importantly improving the adequacy of the samples addressed to molecular chacaterization of mandatory predictive biomarkers in non-small cell lung cancer [6,17,26]. Even in the transbronchial or percutaneous biopsy approach to peripheral lung lesions, there are studies that demonstrate the validity of ROSE in optimizing the procedures [27,28].

The performance of ROSE in thoracic pathology requires close cooperation between cytopathologists and bronchoscopists, possibly guided by a standardized, univocal, prompt and clear procedural language preventing misleading indications, as widely adopted in other settings, namely, breast, thyroid, pancreas and salivary gland diseases [19,20,21,24]. 

In thyroid nodule FNA, Muri et al. [22] compared 1304 cases with ROSE categorization based on the Bethesda system and 3726 cases without ROSE demonstrating that category I (nondiagnostic) and III (indeterminate) were 4.3% in the ROSE cohort and 40% in the group without ROSE. Furthermore, benign (category II) and malignant (category VI) were 91.6% and 56.6% in cohorts with and without ROSE, respectively. The authors concluded that ROSE with a Bethesda categorization significantly increased diagnostic accuracy at both qualitative and quantitative levels, resulting a standard of care for thyroid FNA. Similarly, in a study by Fawcett et al. [23] of 309 FNAs and 101 cases with ROSE, implementation of ROSE decreased the nondiagnostic cases from 41.1% to 23.8%. Procedures performed with ROSE also decreased repeated FNAs from 29.1% to 20.8%, even reducing the number of FNAs per nodule from 1.4 to 1.2 (*p* = 0.04).

Kakkar et al. [24] prospectively investigated the role of the Milan system for reporting salivary gland cytopathology (MSRSGC) on ROSE, comparing the results with the final diagnosis in 60 cases of salivary gland FNAs. The authors reported a correlation of MSRSGC classification during ROSE with the final cytological diagnosis in 58 out of 60 cases (96.7%), and 89% of concordance with the definitive histological diagnosis. Therefore, even in salivary gland lesions, the use of MSRSGC categories with ROSE seems to ensure an adequate diagnostic result with high specificity and sensitivity.

The use of ROSE along with the International Academy of Cytology (IAC) Yokohama System for Reporting Breast Fine Needle Aspiration Biopsy (FNAB) Cytopathology has been proposed in a recent work by Agrawal et al. [25]. The authors evaluated 1147 FNABs, 442 (38.5%) undergoing ROSE and 624 (54.4%) histopathology, demonstrating an overall sensitivity and specificity for identifying in situ and malignant lesions of 99.1% and 99.3%, respectively. Of note, ROSE improved the concordance between cytology and biopsy from 77% to 90%, also significantly reducing inadequate cases (*p* < 0.001). No false positives were observed and 0.7% of false negative cases were recorded. The authors concluded that the integration of ROSE and the IAC Yokohama System for breast cytopathology reporting improved accurate diagnosis of breast lesions, prevented missed diagnoses, standardized a reproducible system for the monitoring and auditing of breast pathology services and improved training at pathology centers.

Other cytology reporting systems have been proposed to enhance the communication between clinicians and pathologists. In lung cancer cytology, Hiroshima et al. [28], on behalf of the Japanese Lung Cancer Society and Japanese Society of Clinical Cytology, proposed a four-tiered cytology reporting system analyzing 90 cases who underwent bronchoscopy. The concordance was fair (k = 0.45) but provided more precise information when compared with three-tiered and five-tiered reporting systems. Even though ROSE was not applied, the study confirmed the helpful role of categorization in improving the communication between clinicians and pathologists and even among different institutions.

Furthermore, Boler et al. [29] investigated the role of the Papanicolaou Society of Cytopathology’s 6-tiered categorical system on 101 consecutive pulmonary CT-guided FNAs, reporting an overall agreement of 71% (k = 0.66), while overall agreement increased to 79.5% (k = 0.74) when considering 5-tiered categories combining “suspicious” and “malignant” cases.

Another advantage of the C1–C5 system is the possibility of ranking the results of ROSE for the purposes of data archiving, evaluation of outcomes and statistical processing. In a previous experience aimed at demonstrating the possibility of assessing the adequacy of ROSE by a trained pulmonologist, we also first evidenced the efficacy of a tiered classification scheme using five diagnostic categories from C1 (inadequate) to C5 (diagnostic), as proposed in other organs [18].

To our knowledge, there are no reports on the use and validation of this classification during routine practice of thoracic pathology by FNA. The current experience is the first attempt to evaluate the C1–C5 categories in lung and mediastinal cytological aspirates on a large and consecutive series of cases.

For the operator performing bronchoscopic or percutaneous needle aspiration, immediate and clear knowledge of the outcomes of the sampling carried out is an essential step to optimize the procedure, also limiting the complications due to unnecessary passes. In the case of a ROSE diagnosis of C1, the pulmonologist is aware that the sampling must be repeated and must try to change the technique or target. However, if C2 or C5 are the judgement of the cytopathologist during the ROSE, the interventional pulmonologist has the certainty that the target is adequately centered and can repeat further passes in the same site to obtain further material for cell block preparation and complete genotyping in the case of NSCLC [6,26].

The indeterminate categories, namely, C3 and C4, are not very frequent in our series (0.61% and 10.39%, respectively) and should lead to a repeat of the sampling procedure in order to obtain additional material to hopefully give a more confident diagnosis. However, of the 237 cases in which ROSE provided a diagnosis of C4 (doubt for malignancy), 134 (56.5%) were confirmed malignant at the final evaluation.

The concordance rate between ROSE and the final diagnosis has been already reported in the literature, mainly for EBUS-TBNA procedures. Of note, in a study by Nakajima et al. on 438 patients undergoing EBUS-TBNA for lung cancer staging, the diagnostic concordance was 94.3% [10]. In addition, Khan et al. [11] evaluated data from 112 patients undergoing EBUS-TBNA disclosing only 1.9% of discordant results. Similarly, Capuena Auledas et al. [12] reported a sensitivity, specificity and overall accuracy of ROSE of 98.6%, 97.2% and 98.5%, respectively, in a study on 637 lymph nodes sampled by EBUS-TBNA. 

Finally, Fassina et al. [13] observed a high accuracy of ROSE even in CT-guided fine needle aspiration of lung nodules, where only three false negative results using ROSE were observed in a series of 311 patients, with a final sensitivity of 96.3% and specificity of 100%.

In our series, the concordance rate between ROSE and the final diagnosis was high, not only in EBUS-TBNA procedures, but also in a transbronchial approach to peripheral pulmonary lesions and in the percutaneous needle aspiration of lung and mediastinal pathologies. Indeed, considering the large number of analyzed patients, the overall rate of false positives and false negatives using ROSE was 1.88% and 3.2%, respectively. The overall concordance between the ROSE categories and the final diagnosis (K = 0.8960) was not statistically different when considering the various type of sampling procedures and target sites. Interestingly, the greatest number of discrepancies occurred between categories C1 and C4 (Table 3 and Figure 2). In total, 28 cases defined as C4 using rapid evaluation were C1 at the final assessment (1.2%), while 30 cases diagnosed as C1 using ROSE were classified as C4 at the final diagnosis (1.3%). These cases, which in any case occurred at a minimal percentage, are likely due to the difficulty in distinguishing reactive bronchial epithelial cells (reactive atypia) from a well-differentiated adenocarcinoma [30].

A further important advantage of this classification scheme is related to the possibility to simplify in a limited number of categories the diagnostic spectrum of lesions using ROSE procedures when trained pulmonologists are involved in preliminary cytological evaluation.

Several studies have demonstrated the helpful role of non-pathologists (in particular pulmonologists) in reliably assessing the adequacy of samples and recognition of neoplastic or granulomatous conditions in ROSE smears. Apart from our seminal experience [18], Umeda et al. [31] reported 88.5%, 83% and 86.4% in terms of sensitivity, specificity and diagnostic accuracy, respectively, in a study of 125 patients undergoing EBUS-TBNA and endobronchial ultrasonography with a guide sheath for peripheral lung nodules with ROSE performed by a trained pulmonologist.

In addition, Natali et al. [32] evidenced in a series of 322 ROSE smears from 162 patients a very good interobserver agreement between pathologists, trained pulmonologists and molecular pathologists in estimating the tumor burden on smeared cytology aimed at molecular profiling of lung cancer from lymphadenopathy or from pulmonary lesions.

Even in this setting, a ROSE categorization may facilitate and standardize the diagnostic communication with bronchoscopists at institutions with staff shortages where a pathologist is not regularly available.

A limitation of the C1–C5 system is that it mainly provides information about the diagnostic value of the sample, and in the era of tailored therapies for lung cancer, the operator should also have information about the adequacy of the sampled material in view of the subsequent tumor genotyping for predictive molecular biomarkers. ROSE may ensure the collection of adequate material for molecular profiling [1]. In a randomized controlled trial that compared EBUS-TBNA performed without and with ROSE, Trisolini et al. showed that the use of ROSE increased the percentage of adequate samples obtained by EBUS-TBNA for molecular profiling [17]. In this study, EBUS-TBNA provided material suitable for complete genotyping in 85.7% of cases, but this value was lower in the non-ROSE group (80.3%) in comparison with the ROSE group (90.8%). 

Specimens classified as C5 during ROSE may contain few neoplastic cells or large amounts of necrosis, leading to an inadequate evaluation for genotyping [6]. This consideration should induce the cytopathological societies to consider a new classification system in the case of lung cancer, which should also include information about the adequacy of the sample aimed at additional molecular determinations. Nevertheless, Ravaioli et al. demonstrated that a careful morphological analysis of ROSE material may establish a more precise histologic type, thus leading to recovery of a higher number of samples for molecular characterization [14].

## 5. Conclusions

In conclusion, the use of a C1–C5 classification during ROSE, validated and largely utilized in the diagnostic procedures for breast, thyroid, pancreas and salivary gland lesions, is a useful tool that also facilitates and standardizes communication in the field of lung and mediastinal pathology. Our data, based on a large series of consecutive cases, demonstrate the high concordance between C1 and C5 categories and the cytological final diagnosis in bronchoscopic and percutaneous approaches to thoracic diseases.

The possibility to include categories that also evaluate the adequacy of the sample for genotyping could be added in the near future to further improve the communication between operators during the biopsy procedures.

## Figures and Tables

**Figure 1 diagnostics-12-02777-f001:**
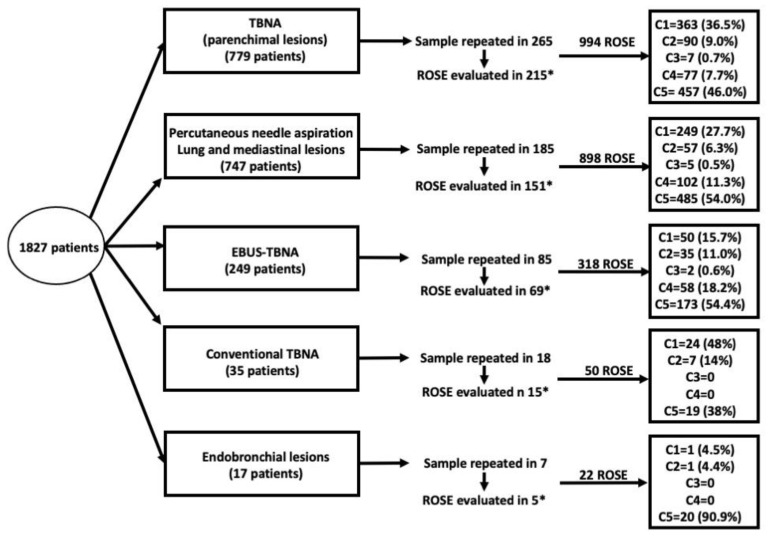
Diagnostic flow with the number of ROSE and C1–C5 categories for each procedure. * ROSE was not performed in 105 patients at the second or third sampling.

**Figure 2 diagnostics-12-02777-f002:**
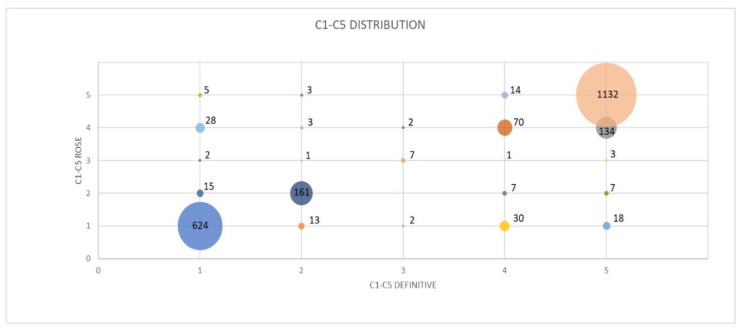
Bubble graph of C1–C5 distribution as evaluated using ROSE and by final cytological diagnosis.

**Table 1 diagnostics-12-02777-t001:** Summary of the procedures performed with ROSE.

Techniques	Number	Percentage (%)
EBUS-TBNA (Lymph nodes)	318	13.9
TBNA (Parenchymal lesions)	994	43.5
Percutaneous	898	39.3
Needle aspiration (Central endobronchial lesions)	22	0.9
Conventional TBNA (Lymph nodes)	50	2.1
Total	2282	

**Table 2 diagnostics-12-02777-t002:** Definitive cytological diagnosis on 2282 samples.

Benign Conditions		
	Number	Percentage (%)
Granulomatous disease	25	1.10
Benign tumors	43	1.88
Reactive lymph node	131	5.74
Inflammation	142	6.22
Other	2	0.09
Malignant lesions		
	Number	Percentage (%)
Adenocarcinoma	566	24.80
Squamous cell carcinoma	116	5.08
Small cell lung cancer	50	2.19
Large cell lung cancer	5	0.22
Non-small cell lung cancer, NOS (not otherwise specified)	443	19.41
Lymphoma	49	2.15
Carcinoid	20	0.88
Metastasis	140	6.13
Thymoma	15	0.66
Other cancer	9	0.39
Non diagnostic samples		
	526	23.05

**Table 3 diagnostics-12-02777-t003:** C1–C5 classification distribution during ROSE and in definitive diagnoses (percentage refers to the incidence of ROSE categories (rows) out of the total number of each single category defined by final cytological diagnosis (columns)).

	C1 Definitive	C2 Definitive	C3 Definitive	C4 Definitive	C5 Definitive	Total
**C1** **ROSE**	624 (92.58%)	13 (7.18%)	2 (18.18%)	30 (24.59%)	18 (1.39%)	687
**C2** **ROSE**	15 (2.23%)	161 (88.95%)	0 (0%)	7 (5.74%)	7 (0.54%)	190
**C3** **ROSE**	2 (0.30%)	1 (0.55%)	7 (63.64%)	1 (0.82%)	3 (0.23%)	14
**C4** **ROSE**	28 (4.15%)	3 (1.66%)	2 (18.18 %)	70 (57.38%)	134 (10.36%)	237
**C5** **ROSE**	5 (0.74%)	3 (1.66%)	0 (0%)	14 (11.48%)	1132 (87.48%)	1154
**Total**	674	181	11	122	1294	2282

**Table 4 diagnostics-12-02777-t004:** Cohen’s kappa for “C1–C5 classification” concordance divided for procedures.

Procedure	Cohen’s Kappa	Standard Error	95% CI
EBUS-TBNA + cTBNA (lymph node)	0.8618	0.0582	0.75–0.97
TBNA (lung)	0.8980	0.0273	0.84–0.95
Percutaneous needle aspiration	0.8552	0.0291	0.80–0.91
Endobronchial needle aspiration	0.8905	0.2025	0.49–1.29

**Table 5 diagnostics-12-02777-t005:** Cohen’s kappa for “C1–C5 classification” divided for organ.

Organ	Cohens’ Kappa	Standard Error	95% CI
Lung	0.8817	0.0211	0.84–0.92
Lymph node	0.8444	0.0547	0.73–0.95
Anterior mediastinum	0.8801	0.0831	0.72–1.04

## Data Availability

The data that support the findings of this study are available from the corresponding author upon reasonable request.

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
