# Peer review of "Validation of a Cytological Classification System for the Rapid On-Site Evaluation (Rose) of Pulmonary and Mediastinal Needle Aspirates"

_diagnostics, 2022, doi:10.3390/diagnostics12112777_

Round 1
Reviewer 1 Report
A well designed and reported review with appropriate references to the main purposes in performing ROSE. A possible suggestion could be to address to the commonest multi-tiered classification systems in respiratory cytology (e.g., Hiroshima et al, Acta Cytologica, 2020; Boler et al, Diagnostic Cytopathology, 2020).
Author Response
Reviewer 1: A well designed and reported review with appropriate references to the main purposes in performing ROSE. A possible suggestion could be to address to the commonest multi-tiered classification systems in respiratory cytology (e.g., Hiroshima et al, Acta Cytologica, 2020; Boler et al, Diagnostic Cytopathology, 2020).
Authors’reply: we thank the reviewer for the favorable comments. We inserted in the discussion section (lines 197-209) a specific comment on the mentioned papers on respiratory cytology, as follows:
Other cytology reporting systems have been proposed to enhance the communication between clinicians and pathologists. In lung cancer cytology, Hiroshima et al. [30] on behalf of the Japanese Lung Cancer Society and Japanese Society of Clinical Cytology proposed a four-tiered cytology reporting system analyzing 90 cases who underwent bronchoscopy. The concordance was fair (κ = 0.45) but provided more precise information when compared with a three-tiered and five-tiered reporting systems.Although not applied on ROSE, the study confirmed the helpful role of categorization in improving the communication between clinicians and pathologists and even among different institutions. Again, Boler et al. [31] investigated the role of the Papanicolaou Society of Cytopathology 6-tiered categorical system on 101 consecutive pulmonary CT-guided FNA reporting an overall agreement of 71% (k= 0.66), while overall agreement increased to 79.5% (k= 0.74) when considering 5-tiered categories combining “suspicious” and “malignant”.
We have also included the mentioned papers in the references (30,31)
Reviewer 2 Report
Dear colleagues, first of all congratulations for your efforts and work. It is an interesting topic with understandable content. The only point that wasn't immediately clear to me was the ''percentage'' addition on the Tables you presented. To what percentages are you referring exactly? Describe it clearly, so it is easy for the reader to understand what you mean. Which parameters are you describing when using the percentage column?
Author Response
Reviewer 2: Dear colleagues, first of all congratulations for your efforts and work. It is an interesting topic with understandable content. The only point that wasn't immediately clear to me was the ''percentage'' addition on the Tables you presented. To what percentages are you referring exactly? Describe it clearly, so it is easy for the reader to understand what you mean. Which parameters are you describing when using the percentage column?
Authors’reply: we thank the reviewer for the favorable comments and for observing the difficulty of interpreting percentage values in Tab 3. Really, we agree that such values in Tab.3 were not clear. We added an explanation in the table legend "(percentage refers to the incidence of ROSE categories (rows) out of the total number of each single category defined by final cytological diagnosis (columns))". Furthermore, we removed the percentage values in the "total" row and column since in fact this values have no meaning.
Reviewer 3 Report
In their retrospective study, the authors have analysed 2282 consecutive ROSE performed on 1827 patients between 2016 and 2020:
· 994 cases of transbronchial needle aspiration (TBNA) in peripheral pulmonary lesions,
· 898 transthoracic FNA,
· 318 ultrasound-guided TBNA,
· 50 conventional TBNA
· 22 endo-bronchial TBNA.
Comparing results obtained by ROSE with the definitive pathological report, the authors have reported a false positive rate of 1.88% and false negatives rate of 3.2%. Sensitivity, specificity, positive and negative predictive values of ROSE were 94.84%, 95.05%, 96.89% and 91.87%, respectively.
After a well conducted analysis the authors have concluded that a classification scheme of ROSE from C1 to C5 during bronchoscopic and percutaneous procedures could improve the clinical management of patients with thoracic lesions.
I have really appreciated this work: it is a well written paper on a well conducted study
· Introduction: offers a precise and concise overview on the issue.
· Methods: well-articulated and accurate
· Results: accurate and detailed presentation
· Discussion: clear and comprehensive
· Tables and figure: interesting and not redundant
I have just two comment:
1. It is clear that only two expert cytopathologist underwent ROSE, but I am also interest in the number of Interventional pneumologist who did the biopsy.
2. I suggest to add a picture with the diagnostic flow, in order to graphically clarified the path and the % of non-diagnostic procedures.
Author Response
Reviewer 3:
Comment.1: It is clear that only two expert cytopathologists underwent ROSE, but I am also interest in the number of Interventional pneumologist who did the biopsy.
Authors’reply:
we thank the reviewer for the favorable comments. According to his comment, we inserted the number of pneumologists (n=5) performing the FNA in the Materials and Method section of the paper.
Comment.2: I suggest to add a picture with the diagnostic flow, in order to graphically clarified the path and the % of non-diagnostic procedures.
Authors’reply:
Thank you for this suggestion, that will improve our paper. We added a Figure (Fig.1) illustrating the diagnostic flow and the percentage of each cytological category (including the non-diagnostic results)